# Epidemiological Aspects of Maternal and Congenital Toxoplasmosis in Panama

**DOI:** 10.3390/pathogens10060764

**Published:** 2021-06-17

**Authors:** Carlos Flores, Delba Villalobos-Cerrud, Jovanna Borace, Lorena Fábrega, Ximena Norero, X. Sáez-Llorens, María Teresa Moreno, Carlos M. Restrepo, Alejandro Llanes, Mario Quijada R., Mayrene Ladrón De Guevara, German Guzmán, Valli de la Guardia, Anabel García, María F. Lucero, Digna Wong, Rima Mcleod, Mariangela Soberon, Zuleima Caballero E.

**Affiliations:** 1Centro de Biología Celular y Molecular de Enfermedades, Instituto de Investigaciones Científicas y Servicios de Alta Tecnología, Asociación de Interés Público (INDICASAT-AIP), Panama City 0843-01103, Panama; cflores@hst.gob.pa (C.F.); delba.villalobos@up.ac.pa (D.V.-C.); lorena.fabrega@up.ac.pa (L.F.); crestrepo@indicasat.org.pa (C.M.R.); allanes@indicasat.org.pa (A.L.); mario.quijada@up.ac.pa (M.Q.R.); anabel.garcia@sydney.edu.au (A.G.); msoberon@indicasat.org.pa (M.S.); 2Laboratorio Clínico, Hospital Santo Tomás, Panama City 0816-00383, Panama; jborace@hst.gob.pa; 3Departamento de Microbiología y Parasitología, Escuela de Biología, Facultad de Ciencias Naturales Exactas y Tecnología, Universidad de Panamá, Panama 4, Panama City 3366, Panama; 4Departamento de Infectología, Hospital del Niño Dr. José Renán Esquivel, Panama City 0816-00383, Panama; ximena.norero@cevaxin.com (X.N.); xsaezll@cwpanama.net (X.S.-L.); mtmsaez@cwpanama.net (M.T.M.); 5Sistema Nacional de Investigación-Secretaría Nacional de Ciencia, Tecnología e Innovasión (SNI-SENACYT), Panama City 0816-02852, Panama; 6Maternidad del Hospital Santo Tomás, Panama City 0816-00383, Panama; mladron@hst.gob.pa (M.L.D.G.); valli.delaguardia@cevaxin.com (V.d.l.G.); 7Facultad de Medicina, Universidad de Panamá, Panama 4, Panama City 3366, Panama; german.guzman@up.ac.pa (G.G.); maria.lucero@up.ac.pa (M.F.L.); 8Centros de Investigaciones Clínicas y Medicina Traslacional, Instituto de Investigaciones Científicas y Servicios de Alta Tecnología, Asociación de Interés Público (INDICASAT-AIP), Panama City 0843-01103, Panama; dwong@indicasat.org.pa; 9Toxoplasmosis Center, University of Chicago Medicine, Chicago, IL 60637, USA; rmcleod@bsd.uchicago.edu

**Keywords:** *Toxoplasma gondii*, seroprevalence, risk factors, pregnant women, newborns, perinatal, maternal and congenital toxoplasmosis

## Abstract

In Panama, epidemiological data on congenital toxoplasmosis are limited, making it difficult to understand the scope of clinical manifestations in the population and factors that may increase the risk of infection. This study provides insight into the epidemiological situation of maternal and congenital toxoplasmosis in Panama and contributing information on the burden of this disease in Central America. Blood samples were collected from 2326 pregnant women and used for the detection of anti-*T. gondii* antibodies. A high seroprevalence (44.41%) was observed for *T. gondii* infection in pregnant women from different regions of Panama, with an estimated incidence rate of congenital toxoplasmosis of 3.8 cases per 1000 live births. The main risk factors associated with *T. gondii* infection using bivariate statistical analysis were an elementary level education and maternal age range of 34-45 years. Multivariate statistical analyses revealed that in some regions (San Miguelito, North and West regions), the number of positive cases correlated with the presence of pets, stray dogs and the consumption of poultry. In other regions (East and Metropolitan regions), the absence of pets was considered a protective factor associated with negative cases, while the presence of stray cats and the age range of 25–34 years did not represent any risk in these regions.

## 1. Introduction

Toxoplasmosis is one of the most common and widespread human parasitic infections worldwide, caused by the Apicomplexan parasite *Toxoplasma gondii*. According to a report from the World Health Organization, congenital toxoplasmosis (CT) is an important global burden of ill health [1]. In 2013, the global incidence of CT was estimated at 190,100 annual cases (95% CI: 179,300–206, 300), with a rate of 1.5 cases per 1000 live births [1]. Despite this significant incidence, CT has not traditionally been considered an important global public health concern, and remains a neglected disease with serious congenital, neurological and ocular sequelae [1]. These facts, together with its high prevalence reported in vulnerable populations with a low socioeconomic level, contribute to perpetuate poverty, mainly in underdeveloped countries [2,3].

Low levels of education and income are socioeconomic factors linked to poor food and hygiene practices, which have been associated with a wide variety of infectious diseases [4,5,6]. In the case of *T. gondii*, these last two factors are known to be crucial in the transmission processes [7,8]. Currently, three main transmission routes have been identified, namely, foodborne, animal-to-human (zoonotic) and mother-to-child (congenital) transmission [9]. Foodborne transmission occurs through the ingestion of food contaminated with tissue cysts or oocysts of the parasite, and is the most frequent, simple and efficient transmission route. [10,11]. The consumption of pork meat is considered to be a concern for the transmission of this parasite in Central and South America [12]. In Panama, a study of several provinces showed high percentages of infection in pork meat [13]. The consumption of contaminated food is possibly the main risk factor in Latin American countries, contributing greatly to the burden of CT in the region, with 1.8 to 3.4 cases per 1000 live newborns [1,10,12]. Nevertheless, this incidence may be underestimated, due to the inadequate interpretation of serological tests and the lack of follow-up of at-risk patients. The frequency of *T. gondii* infection in pregnant women reported in these countries is generally high and varies depending on the region [14]. In Brazil, most of the states have reported percentages higher than 50%, with the highest percentage of 91.7% reported in 2010 for the city of Fortaleza [15,16]. In Colombia and Peru, the prevalence ranges from 28.0–45.8% and from 35.8–97.6%, respectively [17,18,19]. In Central America, few studies have been published, however, reported data have shown relatively high frequencies in Guatemala (55.8%) and Costa Rica (60%) [20,21]. In contrast, some cities of Mexico have shown relatively low percentages of seropositivity, with maximum values around 6% [7,22].

From a clinical point of view, *T. gondii* infection is considered asymptomatic in most individuals. However, individuals with a compromised immune system and pregnant women are currently the most susceptible groups. In the case of pregnant women, a primary infection by *T. gondii* may be fatal to the fetus, without proper diagnosis and treatment [23,24]. The most serious clinical manifestations of a fetal infection occur in the first trimester of pregnancy, during the early stages of fetal development [25]. However, the incidence of CT increases according to the gestational period, which is an important risk factor [25,26]. The dilation of the placenta and immunological and hormonal variations can increase the chances of transplacental transmission [27,28]. Therefore, if the infection occurs in the first trimester of pregnancy, the chances of infection are approximately 25%, increasing to 54% in the second trimester and 65% in the third trimester [26]. In the last trimester, there is a greater possibility that the newborn will be asymptomatic because of the shorter period of infection [25].

In Panama, most studies on human toxoplasmosis were conducted two decades ago. These studies demonstrated the high prevalence of infection in people with different age ranges, from older adults (88.9%) to children (42.5%) [29,30]. It was also observed that the prevalence increased gradually with age and did not differ between residents in urban (58.6%) or rural (57.5%) areas. In addition, the average incidence rate was estimated at 10.25 cases per year [30]. A more recent study carried out in pregnant women reported a seroprevalence of 50% and an incidence rate of congenital toxoplasmosis of two cases per 1000 live births [31]. These results reaffirm the urgent need to perform epidemiological studies in order to update the information periodically, identify difficulties in applying the diagnosis, and to implement prevention and epidemiological surveillance measures in the communities with the highest prevalence. Here, we analyzed the seroprevalence and risk factors associated with *T. gondii* infection in pregnant women and neonates with suspected congenital transmission from West Panama and different regions of Panama Province. Exploratory statistical analyses were also carried out to determine risk factors related to the transmission dynamics typical of each of the studied regions.

## 2. Results

### 2.1. Seroprevalence of T. gondii Infection in Pregnant Women and Newborns

We enrolled 2326 pregnant women from the West Panama and Panama provinces in the present study. A large percentage (87.71%) of these women with more than twenty weeks of gestation had not had any previous serological test for the detection of *T. gondii*, although they had already started prenatal follow-up for other diseases. Only 8.77% had at least one serological test (IgG or IgM) and just 7.39% had both tests. Therefore, the avidity test was not conclusive for diagnosis in most of the suspected cases of CT identified in this study. Only four cases with less than sixteen weeks of gestation and three cases with indeterminate values for the detection of IgM antibodies were ruled out by the avidity test and clinical follow-up (ultrasound and serology). The rest of the pregnant women received treatment until the baby was born.

The global seroprevalence of *T. gondii* infection is 44.41%. Gestational serological screening yielded different percentages of positivity according to the stage of infection. Serological results were classified depending on detected levels of IgG and IgM antibodies as follows: old or chronic infection (IgG + IgM–), probable recent infection (IgG + IgM+), recent infection (IgG– IgM+) and indeterminate cases (IgG + IgM i). Our results show the highest positivity percentages for patients with old or chronic infections (42.60%), with relatively lower positivity percentages for probable recent infections (1.33%), recent infections (0.21%) and indeterminate cases (0.26%) (Table 1). Nine cases of CT were confirmed through the follow-up of neonates from these groups and through serological and molecular diagnosis. The total percentage of CT was 0.38%, with an estimated incidence rate of 3.8 cases per 1000 live births. A relatively high percentage of pregnant women (55.58%) did not show antibodies (IgG– IgM–) against *T. gondii* infection and were therefore considered a risk group. Approximately 0.15% of these patients had a seroconversion, however, this percentage is probably underestimated, because not all pregnant women with a negative result (IgG– IgM–) returned for a second serological test.

The stratified analysis of prevalence by region showed a high percentage of *T. gondii* infection in the regions studied, with minimum and maximum values of 30.31% in the Metro region and 52.50% in the San Miguelito region, respectively. Statistical analyses using the Chi-squared test with a significance level of 0.05 showed statistically significant differences between the Metro region and all the other regions included in the study (San Miguelito (χ² = 40.84, df =2, *p* =0.0001), East (χ² = 34.85, df = 2, *p* = 0.0001), North (χ² = 17.51, df = 2, *p* = 0.0002), Central (χ² = 7.51, df = 2, *p* = 0.023) and West (χ² = 8.18, df = 2, *p* = 0.016)). The San Miguelito region also showed significant differences to the West (χ² = 14.28, df = 2, *p* = 0.0008) and Central (χ² = 7.586, df = 2, *p* = 0.0225) regions, and the East Region with the West region (χ² = 10.27, df = 2, *p* = 0.0059) (Table 1).

### 2.2. Risk Factors Associated with T. gondii Infection

Different categorical variables were analyzed through the Chi-squared test, with a significance level of 0.05, in order to verify their association with *T. gondii* infection. The results showed a possible association between the percentage of positivity and maternal age or level of schooling. The analyses carried out on the total samples studied showed a statistically significant difference between the oldest age range (G3 = 35–44 years) and the youngest age ranges (G1 = 15–24 and G2 = 25–34 years) (G3 vs. G2: χ²= 12.53, df = 2, *p* = 0.0019; G3 vs. G1: χ²= 20.65, df=2, *p* = 0.0001). The stratified analysis of the age data according to geographical area showed a higher percentage of infection in the G3 group in all the studied regions. However, G3 only showed statistically significant differences with the other age groups (G1 and G2) in the Metro (G3 vs. G1: χ² = 7.40, df = 2, *p* = 0.0246; G3 vs. G2: χ² = 6.81, df = 2, *p* = 0.0332), East (G3 vs. G1: χ² = 9.49, df = 2, *p* = 0.0087) and West (G3 vs. G1: χ² = 6.35, df = 2, *p* = 0.0418) regions.

Regarding the level of schooling, a significantly lower percentage of infection was observed in women with a university level education, when compared to those with a high school level education (χ² = 7.27, df = 1, *p* = 0.0070) and an elementary level education (χ²= 20.82, df=1, *p* = 0.0001). The comparison between the elementary and high school level showed a relatively lower level of significance (χ² = 9.25, df = 1, *p* = 0.0023). The results of these analyses by region only showed statistically significant differences between the university level and the elementary (χ² = 8.86, df = 2, *p* = 0.0119) and high school (χ² = 6.93, df = 2, *p* = 0.0312) levels in the West region.

### 2.3. Association between Exposure to a Risk Factor and the Occurrence of the Disease

Relative risk measures, such as the odds ratio and prevalence ratio, were calculated in order to determine the magnitude of the association between *T. gondii* infection and several categorical variables, including age range (G1, G2 and G3), level of education (elementary, high school and university), domestic pets (presence or absence of) and stray animals (presence of dogs or cats). The results showed that some of these categories may be associated with the occurrence of infection. The G3 age range and an elementary school level of education exhibited a strong association with parasite infection in all samples, either when analyzed globally or stratified by region (Figure 1). The youngest age range (G1) was associated with infection in the Central region and the G2 range, in the East and San Miguelito regions. The presence of domestic pets was also associated with infection in the Central and San Miguelito regions. The presence of stray dogs was associated with infection in the Metro, West, Central and East regions, while the presence of stray cats was only associated with infection in the Metro region.

### 2.4. Relationship Patterns between Different Risk Factors and T. gondii Infection

Different categorical variables evaluated through the surveys provided to pregnant women were jointly analyzed using multiple component analysis (MCA) (Figure 2). This analysis showed relationship patterns between different risk factors related to *T. gondii* infection. The variance explained through this analysis for the total population analyzed was 23.3%. The relationship patterns were analyzed using the positive (IgG+) and negative (IgG–) results of the detection of IgG antibodies as reference points. The results obtained showed a close relationship between the positive cases (IgG+) and the San Miguelito, North and West regions. These risk factors in turn were closely related to the presence of pets, stray dogs and the consumption of poultry. The Metro and East regions were more closely related to the presence of stray cats, the G2 age range, the absence of pets and negative cases for IgG– and IgM–.

## 3. Discussion

*T. gondii* infection during pregnancy remains a preventable condition, with treatments available to reduce serious sequelae in the fetus. However, this disease continues to be neglected in most Latin American countries [14,32,33,34]. A recent study conducted in a popular hospital in Panama City showed that 76% of pregnant women had no knowledge about toxoplasmosis [35]. This fact was corroborated in this study when a large percentage (87.71%) of pregnant women with more than twenty weeks of gestation had not had any serological test for the detection of *T. gondii* infection, although they had already had serological screening for other congenital diseases.

Our results also indicated a lower seroprevalence for *T. gondii* infection and a higher incidence rate of CT than those reported in Panama in 2014 [31]. This variation in seroprevalence may be related to the group of pregnant women (high risk pregnancies) analyzed in this study and differences in the sensitivity and specificity indices between the diagnostic tests used in each of the studies. Similarly, the difference observed between incidence rates of CT could be attributed to a variation in the sensitivity index of the diagnostic tests used for the detection of maternal IgM antibodies. These diagnostic tests exhibited sensitivity rates of 92.6% for the Elecsys Toxo IgM assay and 89.9% for the Architect Toxo IgM system in previous validation studies [36,37]. Molecular diagnosis in this study through the PCR-based test was essential for the detection of the parasite and the confirmation of cases of CT. Most of the neonates confirmed by the PCR test did not show IgM antibodies at birth. Detection of postnatal anti-*T. gondii* IgM antibodies is generally complex due to the gestational period in which maternal infection occurred. Some studies have reported a lower sensitivity of diagnostic tests for the detection of IgM antibodies if infection occurs early in pregnancy, whereas the sensitivity tends to be higher if the infection occurs in the final phase of the gestational period [38]. Treatment of patients during pregnancy is another factor that can also decrease the sensitivity of diagnostic tests in the neonate [38,39].

A large percentage of women of childbearing age (55.58%, IgG– IgM–) from Panama City and West Panama are at risk of becoming infected with *T. gondii* and developing an acute infection during pregnancy. Unfortunately, the applicable protocols and regulations for prenatal care are not enforced reliably in Panama and at the appropriate gestational time. The regulations state that serological screening for the detection of IgG and IgM anti-*T. gondii* antibodies must be free of charge, provided in primary care centers (in the first evaluation) and repeated between weeks 29 and 32, if the results of the diagnosis are negative (IgG– IgM–) [40]. However, all pregnant women identified in this study with suspected CT were referred from primary care centers to the high-risk unit of Hospital Santo Tomás for other conditions and not for toxoplasmosis. Furthermore, a large percentage did not have any serological test for *T. gondii*. Failures in serological screening limit the possibilities of determining acute infection using the avidity test, which is efficient only until the sixteenth week of pregnancy. A high avidity result in the second and third trimesters is not conclusive and does not rule out the possibility of infection during the first trimester [41]. These precepts are not clear in the management guidelines for complications in pregnancy established by the Ministry of Health of Panama [42]. Therefore, it is important to carry out a detailed review of the management guidelines and standards for prenatal care. Our findings here highlight the importance of the mandatory reporting of all suspected, confirmed, and fatal cases of CT [43]. We also recommend including mandatory reporting of cases of acute toxoplasmosis infection during pregnancy. These regular reports will be of great help to keep the public databases updated, to identify possible risk factors, to monitor the patients and to organize the acquisition of drugs and laboratory supplies.

Domestic animals have been considered risk factors due to their constant interaction with humans [8,44]. Previous studies carried out in West Panama and in different regions of Panama province showed high frequencies of infection by *T. gondii* in dogs and cats, both strays and pets [45,46]. The bivariate analyses carried out in this study did not show significant association between the presence of domestic animals and the frequency of infection. However, when the relative risk measures (odds ratio and prevalence ratio) were estimated, an association or increased risk was observed between the group of positive pregnant women and the presence of domestic animals (dogs and cats) mainly in the Central region. This region is one of the most densely populated in Panama City, which could increase interactions between its inhabitants and domestic animals [47]. The time of exposure to the parasite also seems to be an important risk factor to be considered in these regions. In our study, the G3 age range showed the highest infection frequencies. However, most of these women were diagnosed with chronic infection (IgG+ IgM–) and only three of them were considered cases of probable recent infection (IgG+ IgM+). Therefore, most of the suspected CT cases belong to the G1 and G2 age ranges, which span from 15 to 34 years old. The results obtained in this study also suggest that communities with higher rates of poverty and unemployment may be associated with higher percentages of seroprevalence. For instance, the San Miguelito and East regions, which showed high percentages of infection, currently have the highest unemployment rates [47,48]. In contrast, the Metro region, which has a higher socioeconomic level, showed a lower percentage of seroprevalence.

*T. gondii* has the ability to use different routes of transmission, due to its complex life cycle. This fact, together with regional habits and customs, generates different relationship patterns capable of influencing the frequency of infection observed in each region. In this study, exploratory statistical analyses revealed a set of potentially important variables for the transmission of the parasite within the regions evaluated. The relationship patterns showed that in the San Miguelito, North and West regions, the presence of pets, stray dogs and the consumption of poultry could be important risk factors associated with the transmission of the parasite. However, the patterns observed in the East and Metro regions seem to be more closely related to the negativity of the infection (IgG– and IgM–). In these regions, the absence of pets was considered a protective factor, and factors such as the G2 age range and the presence of stray cats did not represent a risk for the transmission of the parasite. However, these regions have different urban and demographic characteristics, which may also impact the dynamics of transmission. 

The high prevalence percentage and incidence rates of *T. gondii* observed in this study in pregnant women of childbearing age and their newborns demonstrate how this disease continues to be a latent and underestimated threat in urban regions of Panama Province and West Panama. Our results reflect the neglect in primary care centers and emphasize how the urban, social and cultural characteristics of each region can influence infection dynamics. This study should help direct efforts in epidemiological prevention and surveillance programs. The new data should instigate further research focused on the most relevant risk factors that can affect infection dynamics in Panama and possibly other countries.

## 4. Materials and Methods

### 4.1. The Ethics Statement

The protocols and methodologies used in this study for sample collection, application of surveys and diagnostic tests were reviewed and approved by the Ethics and Research Committee of Hospital Santo Tomás (Panama City, Panama), with protocol approval number 2016-347V3.

### 4.2. Experimental Design

We used a cross-sectional approach to measure the seroprevalence of *T. gondii* infection and risk factors associated with the transmission of the parasite in a population of pregnant women. All pregnant women who attended the high-risk pregnancies section of the maternity ward of Hospital Santo Tomás between June 2017 and December 2018 were included in this study, totaling 2326 patients. This section provides obstetrics services for patients mainly referred by primary healthcare centers. High-risk referrals include simple and major complications of pregnancy, such as high blood glucose levels and alterations of blood pressure. The patients were evaluated through serological methods for the detection of anti-*T. gondii* antibodies (IgG and IgM). Patients with acute *T. gondii* infection received treatment and were followed up until delivery. The newborns with suspected congenital toxoplasmosis were referred to Hospital del Niño (Panama City, Panama), according to standard procedures for medical care in the country, to be diagnosed by serological and molecular tests and clinically evaluated for CT. The definitive diagnosis of CT cases was made after the birth of the newborn by detecting parasite DNA through a polymerase chain reaction (PCR)-based test in samples of cerebrospinal fluid from the newborn and samples of placenta and amniotic fluid from the mother.

### 4.3. Geographical Area Studied 

Pregnant women enrolled in this study reside in different areas of the Panama province, defined as the Central, Metro, East, North and San Miguelito regions. The West region located in the West Panama province was also included. These regions were divided into 45 communities, distributed as follows: six communities from the East region, six from the Central region, five from the Metro region, four from the North region, nine from the San Miguelito region and 15 from the West region (Appendix A). Geographical data related to the percentage of *T. gondii* infection per region were plotted on a map by using the GeoDa software 1.12 (Center for Spatial Data Science, University of Chicago, Chicago, IL, USA) [49] (Figure 3).

### 4.4. Data Collection and Survey Application

Before enrolment in the study, pregnant women received educational talks on the importance of toxoplasmosis diagnosis during pregnancy and relevant information from the study. Patients who agreed to participate in the study received the informed consent document, which was explained by suitably trained personnel. The informed consent was signed by the patients after they had read it carefully. Subsequently, an epidemiological survey was provided, in order to measure different risk factors associated with the transmission of *T. gondii*. The survey included questions related to the patient’s age group (G), defined as 15–24 years old (G1), 25–34 (G2) and 35–44 (G3), region of origin (Central, Metro, West, North, East and San Miguelito), eating and hygiene habits (frequency of street food consumption, food origin, type of meat consumed, consumption of raw meat, frequency of hand washing and frequency of washing fruits and vegetables), interaction with animals (presence or absence of pets and stray animals in the neighborhood), and educational level (elementary school, high school or university) (Appendix A).

### 4.5. Sample Collection

#### 4.5.1. Peripheral Blood

A volume of 5 ml of whole blood was collected from each participant enrolled in the study and placed in a vacutainer tube (Becton, Dickinson and Company, Franklin Lakes, NJ, USA) without an anticoagulant and with a serum separator. Samples were transported in a refrigerated container to Hospital Santo Tomás for diagnosis. Serum was separated from the rest of the blood components by centrifugation at 1600× *g* for 10 minutes at room temperature. Serological diagnostic tests were carried out once the samples arrived at the laboratory. In the case of newborns with suspected CT, a 1 mL blood sample was obtained and placed in microtainer tubes (BD Microtainer, Franklin Lakes, NJ, USA). Peripheral blood samples were drawn via venipuncture.

#### 4.5.2. Placental Samples

Approximately 200 grams of fetal placenta were collected from patients with acute *T. gondii* infection. Samples were taken by the obstetrician in charge of delivery or cesarean section, after expulsion of the placenta. The sample was collected from the region closest to the insertion of the umbilical cord of the newborn and stored in a sterile saline solution with penicillin (100 U/mL) and streptomycin (100 µg/mL). Placental samples were transported in a refrigerated container to INDICASAT AIP and stored at −20 °C.

#### 4.5.3. Cerebrospinal Fluid in the Newborns

To confirm *T. gondii* infection in newborn infants with inconclusive serological diagnosis and clinical signs, a sample of cerebrospinal fluid was taken by the attending neonatologist through lumbar puncture. Approximately 1.5 to 2 ml of cerebrospinal fluid was collected in a sterile tube, which was transported to INDICASAT AIP and stored at −20 °C for subsequent analysis.

### 4.6. Serological and Molecular Diagnosis

#### 4.6.1. Detection of anti-*T. gondii* Antibodies in Serum Samples

The Elecsys Toxo IgG and IgM tests systems (Roche Diagnostics, Mannheim, Germany) were used to detect *T. gondii* antibodies in serum samples collected from pregnant women. This automated system consists of electro-chemiluminescence immunoassays using the sandwich principle for the detection of anti-*T. gondii* IgG and IgM antibodies in serum samples. The Elecsys^®^ Toxo IgG avidity assay was used to distinguish a recent infection from an older one. All commercial tests were performed according to the manufacturer’s instructions and the test results were analyzed and quantified with the cobas e 601 system (Roche Diagnostics, Mannheim, Germany). The serological diagnosis of the newborn was performed through the commercial test routinely used at the Hospital del Niño for the detection of IgM antibodies.

#### 4.6.2. Detection of *T. gondii* Infection in Placental Samples

Placenta samples of patients diagnosed with acute *T. gondii* infection by use of different serological tests (Elecsys Toxo IgG, IgM and avidity assay) were also screened with a PCR-based test. This procedure was carried out according to the methodology previously described for the detection of *T. gondii* DNA using the B1 gene as a molecular marker [50].

#### 4.6.3. Detection of *T. gondii* Infection in Cerebrospinal Fluid Samples from Newborns

Cerebrospinal fluid samples from neonates with suspected congenital toxoplasmosis and inconclusive serological results were analyzed with the PCR-based test, as previously described [50]. 

### 4.7. Statistical Analysis

Basic descriptive statistical measures and prevalence values were calculated using built-in statistical functions of R version 4.0.0 (R Core Team, Vienna, Austria) [51]. The Chi-squared (χ^2^) test with a significance level of 0.05 was used to verify the statistical differences between the prevalence percentages found in each of the analyzed regions. This test was also used to determine the association between the different categorical variables evaluated and the percentages of positivity for *T. gondii* infection. Subsequently, the magnitude of these associations was estimated through relative risk measures, such as odds ratio and prevalence ratio. These analyses were carried out with GraphPad version 6.1 (GraphPad Software Inc., San Diego, CA, USA) [52]. 

A multivariate analysis was carried out to find the most relevant variables related to *T. gondii* infection. The dimensionality of the dataset was reduced using a multiple correspondence analysis (MCA), which is an extension of correspondence analysis (CA) that allows the evaluation of the pattern of relationships of several categorical variables using geometrical methods. Briefly, the data were reduced using the Burt’s method, as implemented in the R package FactoMineR version 1.34, and results were visualized using the Factoextra Package version 1.0.5 [53,54].

## 5. Conclusions

*T. gondii* infection in pregnant women is a public health problem in urban regions of Panama. The high frequency of infection and incidence rates of congenital toxoplasmosis reported in this study revealed the high exposure to the parasite and the effectiveness of the transmission routes (foodborne, zoonotic and congenital) in the regions studied. These results also highlight the deficiencies in the prevention programs and the prenatal follow-up protocol and the inadequate application of diagnostic tests in primary care centers. According to the data obtained in this study, the use of both PCR and serological tests improved parasite detection in neonates with suspected congenital toxoplasmosis. Based on these findings, we recommend the implementation of prevention measures and epidemiological surveillance aimed at pregnant women at risk of infection, these represent a high percentage (55.58%).

## Figures and Tables

**Figure 1 pathogens-10-00764-f001:**
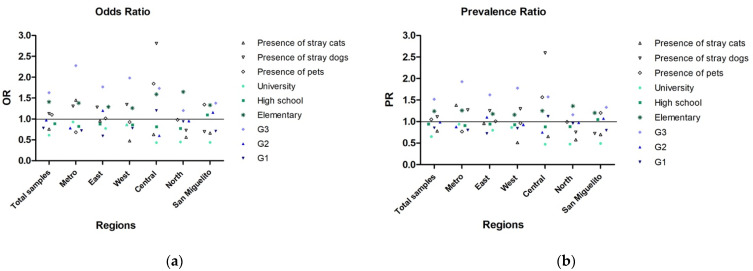
Risk factors associated with *T. gondii* infection by region were estimated through relative risk measures (**a**) odds ratio and (**b**) prevalence ratio, and are shown as follows: presence of animals (pets (dogs and cats) and stray dogs and cats), education level (university, high school and elementary school) and age (G1: 15 -24, G2: 25-34, G3: 35-44). A value of 1.0 indicates no association, a value greater than 1.0 indicates a positive association or increased risk, and a value less than 1.0 indicates an inverse association or decreased risk. The horizontal line delimits the values greater and less than 1.0.

**Figure 2 pathogens-10-00764-f002:**
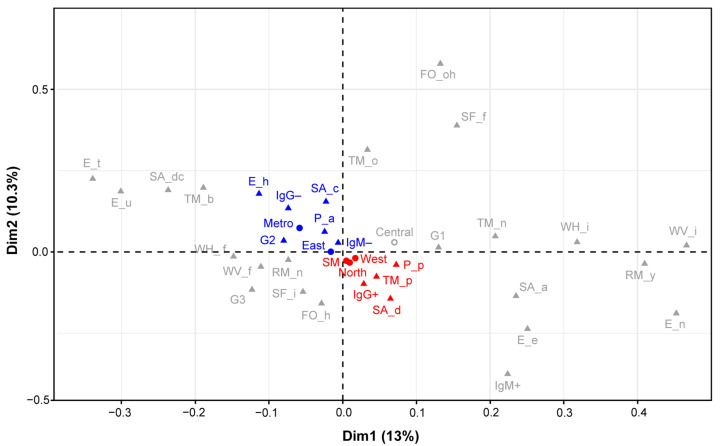
Multiple Correspondence Analysis (MCA). Dimensions 1 and 2 explain 23.3% of the variance of the data set and represent the space where the different categories of variables are expressed. The blue and red dots indicate the categories related to negativity (IgG-) and positivity (IgG +) of the IgG antibody respectively. The most relevant categorical variables are shown as follows: SM (San Miguelito Region), West (West Region), North (North Region), IgG + (positive IgG), SA_d (Stray animal_presence of dog), TM_p (Type of consumed meat_ poultry), P_p (Presence of pets), East (East Region), Metro (Metro Region), IgM- (negative IgM), P_a (Pets_ absence of pets), G2 (Age group 2), IgG- (negative IgG), SA_c (Stray animal_ presence of cats), E_h (Educational level_high school). The other categories can be seen in Appendix A.

**Figure 3 pathogens-10-00764-f003:**
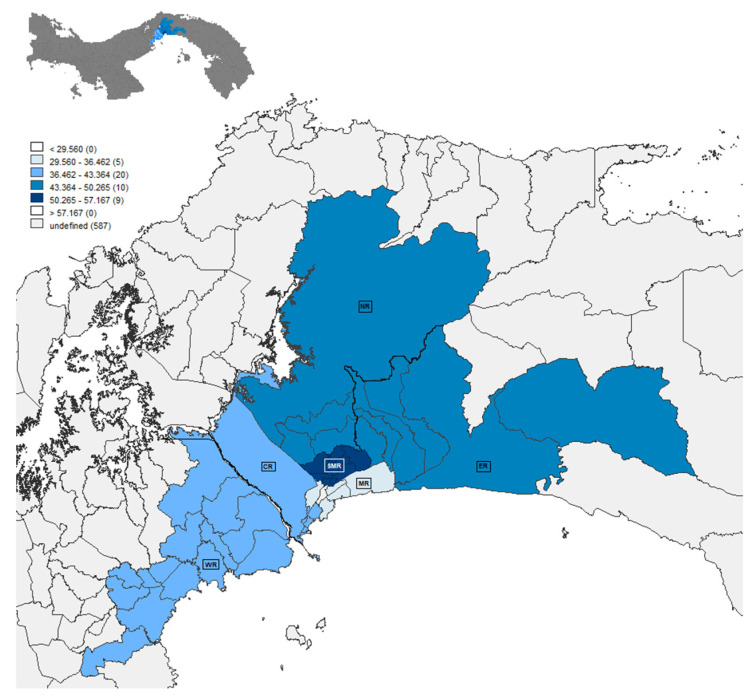
Map of Panama City and West Panama. The evaluated regions were defined with the following abbreviations: SMR = San Miguelito Region, NR = North Region, ER = East Region, WR = West Region, MR = Metro Region and CR = Central Region. The blue scale shows the ranges of the *T. gondii* infection percentages according to the mean ± standard deviation as follows: SM (50.26–57.16), ER and NR (43.36–50.26), CR and WR (36.46–43.36) and MR (29.56–36.46).

**Table 1 pathogens-10-00764-t001:** Seroprevalence of *T. gondii* infection in pregnant women and congenital toxoplasmosis in newborns of West Panama and different regions of Panama Province.

		No. (%), (95% CI)							
				Suspected cases of CT					
Region	No. of Patients	Prevalence	Old Infection IgG+ IgM-	Indeterminatecases IgG+ IgM i	Probable Recent Infection IgG+ IgM+	Recent Infection IgG- IgM+	At-Risk Group G- IgM-	CT Confirmed by PCR	Age ^a^ (Years)
Metro	353	107 (30.31) (25.62–35.44)	100 (28.32) (23.74–33.38)	1 (0.28) (0.15–1.82)	5 (1.42) (0.52–3.47)	1 (0.28) (0.01–1.82)	246 (69.68) (64.56–74.38)	0 (0) (0–1.34)	27.83 ± 0.33
West	444	178 (40.09) (35.52–44.83)	171 (38.51) (33.99–43.23)	1 (0.23) (0.01–1.45)	6 (1.35) (0.55–3.07)	0 (0) (0–1.07)	266 (59.91 (55.17–64.47)	0(0) (0–1.07)	27.73 ± 0.29
Central	227	94 (41.41) (34.98–48.13)	90 (39.65) (33.30–46.35)	0 (0) (0–2.07)	4 (1.76) (0.56–4.75)	0 (0) (0–2.07)	133 (58.59) (51.87–65.01)	1 (0.44) (0.02–2.81)	27.34 ± 0.44
North	296	137 (46.28) (40.52–52.14)	133 (44.93) (39.20–50.80)	2 (0.68) (0.12–2.69)	2 (0.68) (0.12–2.69)	0 (0) (0–1.60)	159 (53.72) (47.86–59.48)	0 (0) (0–1.60)	26.53 ± 0.36
East	526	265 (50.38) (46.02–54.73)	253 (48.09) (43.76–52.46)	2 (0.38) (0.06–1.52)	8 (1.52) (0.71–3.10)	2 (0.38) (0.06–1.52)	261 (49.62) (45.27–53.97)	3 (0.57) (0.15–1.80)	26.65 ± 0.27
San Miguelito	480	252 (52.50) (47.93–57.03)	244 (50.83) (46.27–55.38)	0 (0) (0–0.99)	6 (1.25) (0.51–2.84)	2 (0.42) (0.07–1.66)	228 (47.50) (42.97–52.07)	5 (1.04) (0.38–2.56)	27.61 ± 0.27
**Total**	**2326**	**1033 (44.41)** **(42.38–46.46)**	**991 (42.60)** **(40.59–44.65)**	**6 (0.26)** **(0.10–0.59)**	**31 (1.33)** **(0.92–1.91)**	**5 (0.21)** **(0.08–0.53)**	**1293 (55.58)** **(53.54–57.62)**	**9 (0.38)** **(0.19–0.76)**	**27.28 ± 0.13**

^a^ Age is shown as mean ± standard error of the mean (S.E.M.). **CI:** confidence interval; **CT:** congenital toxoplasmosis; **IgM i:** indeterminate IgM antibody; No.: number.

## Data Availability

Data supporting the findings of this study are available on figshare under the digital object identifier (DOI) https://doi.org/10.6084/m9.figshare.14392838, accessed on 15 December 2020.

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
