# Peer review of "Epidemiological Aspects of Maternal and Congenital Toxoplasmosis in Panama"

_pathogens, 2021, doi:10.3390/pathogens10060764_

Round 1

Reviewer 1 Report

I don’t appreciate the OR quantification using an overlay histogram. As an example, an important role as risk factor seems to be played by pets. However, association between exposure to this risk factor and the occurrence of the disease sounds to me less important than other factors, with the exception of the central region. 

Author Response

Reviewer 1

Dear reviewer, thank you very much for your comments.

  1. We have changed the presentation of this graph to improve the visualization of the association values (odds ratio), please see the attachment. Check if these values can be better understood in this way. Due to your comments we also perceive an error in figure 1b with the term risk ratio. In the case of our study (cross-sectional study), it is more appropriate to use the term prevalence ratio. The formula used to calculate both ratios (prevalence ratio and risk ratio) is the same. Therefore, we only change the term risk to the term prevalence in lines 172 of the results section, in the figure and legend line 186, in the discussion section line 265 and in the materials and methods section line 410. Thank you for your very precise comments.

Reviewer 2 Report

The manuscript provided very valuable data about maternal and congenital toxoplasmosis in Panama which may be interesting for epidemiologist. I enjoyed reading this manuscript, it was well written and informative and clearly adds valuable information to the field of parasitology.
Futhermore, the examination were conducted on a large number of samples which increased the value of the article.
In my opinion the manuscript is well written and except for 3 minor issues I have no comments.

  1. You have a native English speaker go over the manuscript just to help and improve tenses and syntax right. The English is generally very good, it just needs some tweaking.
  2. The full parasites names should be italic, especially in the part of M&M (line 381; line 392; line 397; line )
  3. I have problem with line 418-419. In my opinion this sentence is not necessary.

Author Response

Reviewer 2

Dear reviewer, thank you very much for your comments.

  1. The text was reviewed and corrected by an English language professional and native speaker.
  2. The names of the parasites were changed to italic in lines 381, 392 and 397 and were verified throughout the manuscript.
  3. Lines 418 and 419 were removed. Those lines are from the journal template, those lines should have been removed. Thank you for perceiving that error.
